# Synonyms and Symptoms of COVID-19 and Individual and Official Actions against the Disease—A Brief Online Survey 6 Months into the Pandemic and on the Threshold of the Second Wave in Germany

**DOI:** 10.3390/ijerph19010169

**Published:** 2021-12-24

**Authors:** Katharina Auth, Sabine Bohnet, Cornelius Borck, Daniel Drömann, Klaas F. Franzen

**Affiliations:** 1Medical Clinic III, Campus Lübeck, University Hospital Schleswig-Holstein, 23562 Luebeck, Germany; katharina.auth@student.uni-luebeck.de (K.A.); sabine.bohnet@uksh.de (S.B.); daniel.droemann@uksh.de (D.D.); 2Airway Research Center North, Member of the German Center for Lung Research (DZL), 22927 Großhansdorf, Germany; 3Institute for History of Medicine and Science Studies, University of Lübeck, 23562 Luebeck, Germany; Cornelius.borck@uni-luebeck.de

**Keywords:** COVID-19, symptoms, Germany, second wave, online survey

## Abstract

To control the ongoing global pandemic due to SARS-CoV-2, we need to influence people’s behavior. To do so, we require information on people’s knowledge and perception of the disease and their opinions about the importance of containment measures. Therefore, in August 2020, we conducted an anonymous cross-sectional online survey on these topics in 913 participants in Germany. Participants completed a questionnaire on various synonyms and symptoms of corona virus and specified the importance they attributed to individual and regulatory measures. The virus was linked more closely with most synonyms and the discovery in China than with the places of the first larger European outbreaks. General (cold-like) symptoms, such as “cough” and “fever,” were more widely known than COVID-19-specific ones, e.g., “loss of taste and smell.” The widely promoted individual measures “distancing,” “hygiene,” and “(facial) mask wearing” were rated as highly important, as were the corresponding official measures, e.g., the “distancing rule” and “mask mandate.” However, the “corona warning app” and a “vaccine mandate” were rated as less important. A subgroup analysis showed broad agreement between the subgroups on nearly all issues. In conclusion, the survey provided information about the German population’s perception and knowledge of the coronavirus five months into the pandemic; however, participants were younger and more educated than a representative sample. To learn from the beginning and still ongoing pandemic and develop concepts for the future, we need more conclusive studies, especially on the acceptance of further specified lockdowns, the population’s willingness to be vaccinated, and the influence of misinformation on public opinion.

## 1. Introduction

On 11 March 2020, the World Health Organization (WHO) declared the disease caused by severe acute respiratory syndrome coronavirus 2 (SARS-CoV-2), coronavirus disease (COVID-19), a global pandemic [1]. At the time of writing this manuscript, the disease is affecting more than 200 countries [2]. The virus was discovered in Wuhan, China, in December 2019 [1,3]. Initially, the number of infections and deaths rose sharply, and the number of confirmed cases increased in waves to over 108,006,680 infections and more than 2,378,115 reported deaths from or with COVID-19 globally (status on 14 February 2021) [4,5]. In Germany, the disease started to spread rapidly in the middle of March 2020, and the number of new infections and deaths rose dramatically. However, measures taken in the subsequent months enabled the German authorities and government to “control” the spread of the virus and consequently to reduce the number of cases and deaths.

Behavior in populations is known to be influenced by and change because of the population’s knowledge, beliefs, and opinions about a disease [6]. In the case of COVID-19, public opinion was also found to be influenced by the spread of dis- and misinformation about the disease [4,7]. As one of several key elements, the course of the COVID-19 epidemic in Germany depended on the population’s behavior during the lockdown. However, in Germany the “unorthodox thinker” (“Querdenker”) movement and conspiracy theorists started to have an increasing influence on public opinion during the protests against government measures and restrictions. As cases in Germany started to increase again in July and August 2020 to numbers that in mid-October 2020 exceeded those of the first wave in March and April 2020, controlling the spread of the virus became even more important, and after further increases, the government started to discuss a second shutdown or lockdown [8,9]. This second wave increased the need to obtain a picture of the population’s knowledge and opinions about COVID-19 as a basis for understanding and influencing their behavior.

In the case of a rapidly spreading infectious disease such as COVID-19, alternative and additional methods must be used when performing surveys and acquiring information. For example, surveys must be designed to help identify complementary responses and actions by public health authorities and governments. Neither population-representative household surveys nor telephone surveys provide adequate answers in situations with rapidly changing incidences because the preparation and data collection phases of these types of surveys take a long time [10]. Furthermore, telephone surveys have a low response rate [4,11], and both household and telephone surveys require statistical corrections to adjust for nonresponse [12]. In addition, these surveys require a large number of staff. As a possible solution to these problems, online surveys were introduced as a (new) source of data and a feasible alternative because they can reach a large number of respondents in a short time and thus enable researchers to assess a population’s knowledge and perceptions about a disease widely and quickly. Such an approach is particularly important in the case of a rapidly spreading infectious disease such as COVID-19 [13].

Therefore, this study aimed to examine the German population’s perception of the typical symptoms of COVID-19 and the pandemic situation and also to assess which individual and official measures they accept and consider to be important to prevent infection or contain the coronavirus pandemic. The answers to these questions were still unclear when the incidence of COVID-19 started to increase again at the start of the second wave in Germany and are of continued interest as this wave continues. For the reasons discussed above, this study used an online survey as the assessment tool to keep pace with the rapid epidemiological development of COVID-19.

## 2. Materials and Methods

### 2.1. Participants

This study collected data with a cross-sectional online survey. Participants were recruited via the research platform Prolific, which is hosted and managed by Prolific Academic Ltd. (Oxford, UK). Individuals from all over the world can participate in online research studies on the Prolific online platform [4,14], and the platform has 139,403 registered individuals. The majority of these individuals live in the United Kingdom (36%) and the United States (31%), but 1.8% list Germany as their country of residence [15]. Participants in surveys on the Prolific platform receive incentives for completing surveys. For this study, the participants were paid £8.07 (US$10.57) per hour, which resulted in a cost of £1.48 (US$1.94) per survey and an average remuneration of £10.97 (US$14.37) per hour per participant [16]. To focus this study on Germany, we selected participants among registered individuals on the Prolific platform by using a custom prescreening with the filters “nationality: Germany” and “first language: German.” In total, 1594 registered individuals on the Prolific platform were eligible for this study. We aimed to obtain a total sample size of 750 German participants.

In addition to recruiting participants on the Prolific platform, we acquired additional participants by sending the link to the questionnaire to all recipients on the e-mailing list of the University of Luebeck. As an incentive, we offered a raffle of 5 × €5 (US$5.85) vouchers per 100 participants [17]. To detect double participation, we used cookies and specific enrollment information.

The local ethics committee (University of Lübeck) approved the study (AZ 20-290 and date of approval 17 July 2020). The study was conducted according to the Declaration of Helsinki, and the Checklist for Reporting Results of Internet E-Surveys (the CHERRIES statement) [18] was considered during implementation.

### 2.2. Data Collection

The first author, one of the co-authors and the senior author created the questions for the survey, assessed the design and checked the feasibility and validity of each question. The final version of the survey was edited and released by the local ethics committee. The data were collected with a questionnaire, and the online portal “umfrageonline.de” was used to create the electronic version of the questionnaire. Subsequently, the questionnaire was uploaded to the Prolific platform. To avoid interface problems, we completely embedded “umfrageonline.de” in the development process and in the Prolific platform. The participants had to answer each question before they could proceed to the next question and were required to complete the whole questionnaire to receive the incentive.

The questionnaire was activated from 6 August to 17 August 2020. Initially, it was accessible to Prolific users. 750 complete data records were reached via the platform by 10 August on a first come, first served basis (a total of 760 participants were recruited via the Prolific Platform). The questionnaire was then made accessible to recipients of the university’s e-mailing list from 10 August to 17 August. This resulted in an additional 153 data records.

In addition to demographic information, such as gender, age, and level of education, we collected information on COVID-19-related medical history. Additional questions addressed the degree of familiarity with and awareness of terms and symptoms associated with the coronavirus and the importance of various individual and regulatory measures to prevent SARS-CoV-2 infections in Germany. For each question, 7 to 10 relevant terms were listed, and participants were asked to rate the terms on a 6-point rating scale ranging from 1 (−−−), completely unknown or unimportant, to 6 (+++), very well-known or very important.

In addition, to increase the feasibility and validity of the survey the questionnaire included 2 attention check questions (i.e., control questions). These questions had no reference to the topic of the coronavirus, and each had only one clearly correct answer. These 2 questions were used to ensure that participants were paying attention and to avoid random clicking patterns.

### 2.3. Statistical Analysis

To evaluate the rating scale, we determined cut-off values in advance. By using the questionnaire rating scale of 1 to 6, we determined 4.51 as the cut-off value for identifying clearly positive answers. Terms scoring a mean value greater than or equal to 4.51 were classified as known/familiar or important, and terms with a mean value of less than 4.51, as less known/familiar or less important. For subgroup analyses, total scores were calculated according to gender, age, and level of education. The two age groups were created by using the cutoff determined by the median age of 27 years, and the level of education was defined as academic or non-academic

To ensure the security and above all the quality of the data, we created various criteria for including or excluding the questionnaires. First, only completed questionnaires were included in the data analysis; 23 of the 913 surveys had to be excluded because they were incomplete or inconsistent. Next, in accordance with the publication by Geldsetzer [4] all questionnaires with a response time of less than 3 min were excluded from the evaluation; this criterion applied to 2 questionnaires, which were therefore excluded. In a third step, the surveys containing at least one incorrect answer to 1 of the 2 control (attention-check) questions were excluded from the analysis. A total of 25 surveys had one or more incorrectly answered control questions and were excluded.

The data were exported from “umfrageonline.de” in a tabulated format and then formatted and analyzed in Excel. Furthermore, the data were exported to SPSS (IBM Corp., Armonk, NY, USA) and Chi-squared tests and a logistic regression analysis was performed. For the logistic regression analysis ratings from 1 to 3 were classified as unknown/disapproval and ratings from 4 to 6 as known/approval. The independent variables age, gender (female, male, diverse), education (non-academic and academic education), SARS-CoV-2 infection and having been quarantined were analyzed singly and combined for significant correlation with multiple dependent variables (risk assessment, different symptoms of a coronavirus infection, individual and official measures). The results of the logistic regression analysis as odds ratios are only reported if the omnibus test has previously shown significance for the entire model. Graphs were created, edited and prepared for publishing with GraphPad Prism 5.0 (GraphPad Software Inc., San Diego, CA, USA).

## 3. Results

A total of 913 German citizens with a mean age of 29 years (Figure 1a) participated in the anonymous survey in the first half of August 2020 (Figure 1b). After excluding incomplete surveys and surveys with incorrectly answered control questions, 863 (94.5%) surveys were available for analysis.

### 3.1. Characteristics and COVID-19-Related Medical History

The majority of the people surveyed (86%) had an academic education (Abitur [high school diploma] or bachelor’s or master’s degree). A total of 45% of the participants were female; 54.2%, male; and 1%, diverse (Table 1).

Eight participants (1%) reported having been infected with the coronavirus, and 112 respondents (13%) had been in quarantine at some point since the start of the corona pandemic. In response to the question about their self-rated individual risk assessment, 12% counted themselves as part of a risk group, and 6% did not know whether they belonged to a group at higher risk from the coronavirus (Table 2).

### 3.2. Synonyms, Symptoms, and Actions

Several synonyms for the term coronavirus, such as “COVID-19” and “coronavirus,” were rated as very well known, as were the terms “China” and “Wuhan.” Less known were the terms “Ischgl” and “Lombardy” (Figure 2a).

The best-known symptoms of COVID-19 were “cough,” “fever,” “shortness of breath,” “respiratory failure,” and “pneumonia.” “Loss of taste” and “loss of smell” were less well known (Figure 2b).

The individual actions rated most important to prevent a SARS-CoV-2 infection were “distance” (in the sense of keeping physically distanced) and “hygiene,” both of which showed a fairly low standard deviation, followed by “social distancing” (an English term not commonly used in German before the pandemic. In Germany physical contact of shaking hands as a form of polite greeting and hugging among friends is rather common.), two variations of the German word “(facial) mask” (“Mundschutz” and “Maske”), and “vaccine.” Among other actions, the “corona warning app” was rated as less important (Figure 3a).

Many of the listed official measures, such as “distance rule,” “mask mandate,” “contact restrictions,” and “quarantine,” were considered important for containing the coronavirus pandemic in Germany. Only “vaccine mandate” was viewed as less important, but the respective responses had a comparatively high standard deviation (Figure 3b).

### 3.3. Subgroup Analysis

Data were further analyzed by dividing the participants into subgroups on the basis of gender (female, male, diverse), age (27 or younger and 28 years or older), and education (academic and non-academic education).

In all but two of the subgroups, 1% of the participants had a history of coronavirus infection; the exceptions were the non-academic education group in which no-one had been infected, and the diverse gender group in which one out of eight had tested positive for the coronavirus. The percentage of people who had been quarantined at some time during the coronavirus pandemic was 12% in men, 13% in participants with an academic education and in both age groups, 14% in women, 15% in participants with a non-academic education, and 25% in the diverse subgroup (this subgroup comprised a total of only eight participants). In the responses to the question whether participants saw themselves as part of a group at risk from the coronavirus, the gender distribution was almost equal (11% of men and 13% of both women and diverse individuals). In the education subgroups, 11% of the participants with an academic education and 16% of those with a non-academic education counted themselves as part of a risk group. The biggest difference was found between the two age groups: 9% of the participants aged 27 years or younger considered themselves to be part of a group at risk of infection versus 15% of the participants 28 years or older.

Most of the synonyms and terms related to the coronavirus showed similar mean values among all subgroups, and only the mean values of some locations associated with the virus showed a difference from the values in the whole group. “Lombardy” reached a higher mean value in the group of participants aged 28 years or older (3.26 vs. 2.93 in the whole group), whereas in the group with a non-academic education the terms “Lombardy” and “Ischgl” both reached a lower mean value (2.5 vs. 2.93 in the whole group and 3.27 vs. 3.7 respectively). “Wuhan” was rated as less known/familiar in the group of participants with a non-academic education, reaching a mean value of 4.42.

The COVID-19 symptom “loss of smell” was less well known in the non-academic education group than in the whole group (mean, 4.46 vs. 4.86, respectively).

Men and participants aged 28 years or older rated “facial mask (protection class FFP3)” as more important (mean value ≥ 4.51) than he whole group did, whereas participants aged 27 years or younger and the subgroup with a non-academic education rated “facial mask (protection class FFP2)” as less important (mean value < 4.51).

“Ausgangsbeschränkung,” which can be loosely translated as “stay at home order,” was rated as less important by men, participants aged 28 years or older, and the non-academic education subgroup (mean value < 4.51) than by the whole group.

In general, women showed a slightly higher mean value than the overall group for importance/knowledge for nearly all terms and measures and men showed a slightly lower mean value. The largest differences were seen in the diverse group, but these results were not given further consideration because of the small number of participants in this subgroup (*n* = 8).

Additionally, Chi-squared tests showed a significant difference between expected and observed frequency concerning educational level and the variables “individual risk assessment”, the knowledge of “loss of smell” as a symptom of a coronavirus infection and the importance of the “corona warning app” (Pearson’s Chi-squared test *p* = 0, *p* = 0.016, *p* = 0.015). A significant difference could also be observed for quarantining and the official measure “mask mandate” (Fisher’s exact test *p* = 0.033). In the logistic regression analysis, a significant correlation was detected for “corona warning app” as a function of education (odd’s ratio 0.621) and quarantine (odd’s ratio 1.124). For the terms “loss of smell”, “loss of taste” and “lockdown” a combination of the independent variables education (odd’s ratio 0.583), age (odd’s ratio 1.023), gender (odd’s ratio 0.001), SARS-CoV-2 infection (odd’s ratio 0.001) and having been quarantined (odd’s ratio 1.449) showed a significant correlation. “Vaccine mandate” and “mask mandate” showed a significant correlation to the independent variable of quarantine (odd’s ratio 0.637 for vaccine mandate and odd’s ratio 0.16 for mask mandate), for “mask mandate” the combination of education (odd’s ratio 1.581) and quarantine (odd’s ratio 0.637) also was significant. No significant difference between null model and models with variables was found for “cough”, “common cold”, “distance”, “facial mask” (Maske), “facial mask” (Mundschutz), “social distancing”, and “distance rule”.

## 4. Discussion

### 4.1. Discussion

This study examined after five months the extent to which the German population is familiar with the typical symptoms of COVID-19 and the pandemic situation and also to assess which individual and official measures they accept and consider to be important to prevent infection or contain the coronavirus pandemic. Compared with the general public in Germany, the study population had a lower mean age and a higher percentage of people with an academic education. Nevertheless, at the time of writing people younger than 60 years make up the largest share of infected people in Germany [19]. The percentage of study participants who had been infected with SARS-CoV-2 was 0.9%, which was higher than the average of approximately 0.3% of the German population infected at the time of the survey [8,9,20].

In addition to various synonyms, the majority of participants associated the coronavirus more strongly with the place of its first appearance, i.e., China or Wuhan, than with the first European hotspots, i.e., Lombardy or Ischgl, even though the media showed emotional images from both these European locations. The awareness of COVID-19 symptoms showed a clear focus on the lungs as the affected organ. On the one hand, this finding could be attributed to the existing lack of ventilators/intensive care beds, which was communicated especially at the beginning of the pandemic. On the other hand, it could also be explained by the association of coronavirus symptoms with the typical symptoms of influenza. In contrast, the more SARS-CoV-2-specific symptoms loss of smell and loss of taste were less well known. Other COVID-19 symptoms listed by the Robert Koch Institute were substantially less well known because they were rather unspecific cold symptoms. In connection with the increasing number of infections at the time the survey was conducted, this result could indicate that many people in the general population do not realize they are infected with SARS-CoV-2. Consequently, people may continue to have contact with others despite being infectious and not be tested or put themselves in quarantine or isolation.

Distancing, hygiene, and (facial) masks (“Mundschutz”) were rated as particularly important individual measures for fighting the coronavirus, and they correspond to the measures promoted by the Federal Ministry of Health with the formula “AHA” (Abstand, meaning physical distancing; Hygiene, meaning hygiene; and Alltagsmasken, meaning face masks) [21]. However, it remains to be seen whether the survey results reflect the success of this campaign or whether other factors, such as legally stipulated measures, also play a role. The Federal Government’s corona warning app was considered less important, which is also reflected in the current download figures of 17.8 million in a population of 60.74 million smartphone users [8,9,22]. Because the survey’s participants were on average younger and better educated than the general population, they may have been more likely to be familiar with and use devices such as smartphones and tablets. In the general public, the corona warning app might play an even less important role than the numbers in our study indicate. The official measures listed in our questionnaire met with an overall high level of acceptance among the respondents. Only a mandatory vaccination was viewed as less important, whereby the comparatively high standard deviation indicated how opinions diverged on this particular topic.

In the various subgroups, the most striking difference was seen between the risk assessments of the two age groups. The finding that a higher percentage of participants aged 28 years and older counted themselves as part of a group at risk from the coronavirus is probably because age is seen as one of the risk factors for COVID-19, besides the fact that old people having more pre-existing conditions in general. One explanation for differences in the mean values between the subgroups concerning early coronavirus “hotspots” outside of Germany could be varying interest in foreign events and travel among the subgroups.

The significant correlation shown between the independent variable education and the variables “corona warning app”, “risk assessment”, and “loss of smell” might be caused by a better understanding of and a higher interest in the disease, its typical symptoms, (medical) consequences, and measures to prevent the infection among participants with an academic education in comparison to those with a non-academic education. A significant correlation between the experience of quarantine and the approval of a mask mandate and a vaccine mandate could be explained with the restrictions experienced during quarantine (in most cases because of contact tracing, without being infected themselves). This might cause a higher acceptance of mandates for this subgroup in comparison to other factors, e.g., a SARS-CoV-2 infection itself, in order to prevent the spread of the disease and lowering the possibility of another quarantine in the future.

Overall, one must note that the survey was taken in August 2020, a time when the coronavirus pandemic seemed to be under fairly good control in Germany and Europe in general. We do not know whether and how this circumstance might have influenced the participants’ answers and whether we would have obtained different results at the beginning of the pandemic, during the first, second or third wave of infections, or during times of shutdown or lockdown.

### 4.2. Limitations

This study has several limitations. First, the survey was not representative because of the age and educational level of the participants. The majority of participants were aged between 20 and 30 years and had a higher education status. Additionally, the participants were recruited online with the help of a foreign survey platform and the university’s e-mailing list, creating a bias towards younger, more technologically savvy, and—in the case of the e-mailing list—more highly educated individuals. However, at the time of data collection the percentage of respondents who reported having already been infected with SARS-CoV-2 was higher than in the general population. Thus, the missing age cohorts and the higher proportion of people with an academic education may have become less relevant in the data analysis and discussion. A further weakness of the survey is the self-assessment of risk, which did not include questions on previous illnesses to validate the plausibility of the participants’ assessments. Because the questionnaire was based on evaluations and subjective information, only those respondents who answered the attention-check questions incorrectly or gave inconclusive or obviously incorrect answers to the questions were removed from the analysis; the other participants were considered authentic. Further plausibility criteria were difficult to create because of the aim of the study and the rapid development with the accompanying media reporting.

## 5. Conclusions

In conclusion, this survey provides information on the perception and, above all, the knowledge about coronavirus among the German population, although the interviewed cohort was likely to be younger and more educated than a representative sample. Nevertheless, this survey shows that in time-critical situations rapid online surveys can be implemented quickly and easily in Germany, as has been shown for the US and the UK [4]. Based on our results and results of further surveys the authorities should launch additional awareness campaigns. Our findings indicate that the topic of vaccination should had been brought to the forefront to prevent the current stalling of the vaccination campaign in advance, in particular with respect to the aim to achieve herd immunity and thus finally contain COVID-19. The widely advertised corona warning app played a subordinate role among the respondents; however, it offers support for individual contact tracing, which can be particularly important in periods of increasing numbers of cases, e.g., on the threshold of the second wave of infections in Germany. When health authorities are confronted with an excessive number of demands, such apps can play a decisive role in individual risk management and behavior. The evaluation and, above all, the acceptance of new shutdowns or lockdowns after the survey period should be addressed in further surveys. In addition, further studies should also focus more closely on vaccination and the population’s associated prejudices, assessment of false claims, and willingness to be vaccinated.

## Figures and Tables

**Figure 1 ijerph-19-00169-f001:**
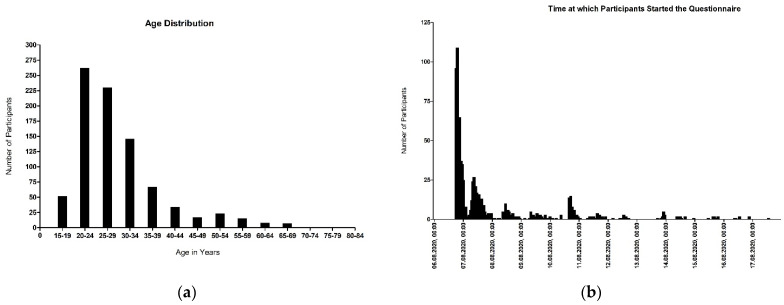
(**a**) Age distribution of participants; (**b**) Date and time when participants started the questionnaire.

**Figure 2 ijerph-19-00169-f002:**
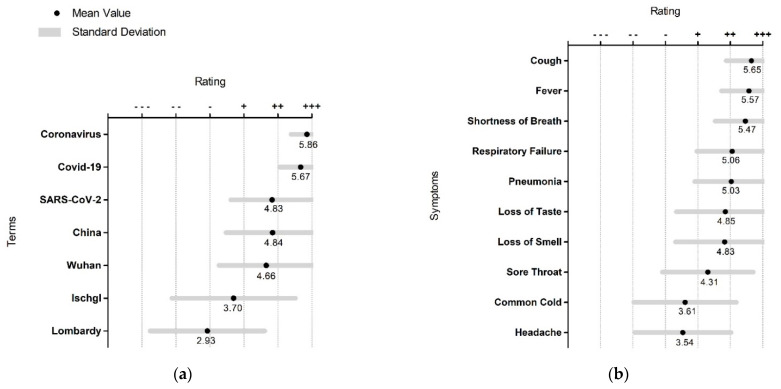
(**a**) Level of familiarity with synonyms and terms related to the coronavirus; (**b**) level of familiarity with COVID-19 symptoms; (ranging from (−−−) for completely unknown to (+++) for very well-known).

**Figure 3 ijerph-19-00169-f003:**
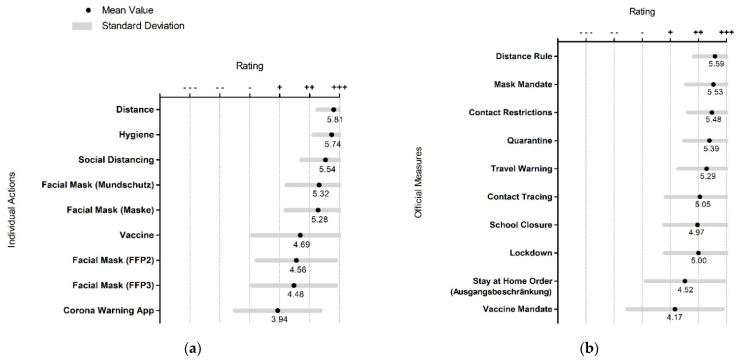
(**a**) Importance of individual actions to prevent SARS-CoV-2 infections; (**b**) importance of official measures for containing the coronavirus pandemic in Germany; (ranging from (−−−) for completely unimportant to (+++) for very important).

**Table 1 ijerph-19-00169-t001:** Participant characteristics (N = 863 after excluding ineligible participants).

Characteristic	
Gender	387 female, 468 male, 8 diverse
Age mean ± SD, y	29 ± 9.8
Age, median, y	27
≤27 y, *n*	467
>27 y, *n*	396
Education	
Academic education, *n*	739
Non-academic education, *n*	124

**Table 2 ijerph-19-00169-t002:** COVID-19-related medical history (N = 863 after excluding ineligible participants).

Medical History	*n*
SARS-CoV-2 infection	8
Quarantined	112
Risk group (self-assessed)	102; 50 additional participants unsure

## Data Availability

All relevant data are presented in this paper, and the raw data presented in this study are available on request from the corresponding author.

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
