# Peer review of "Synonyms and Symptoms of COVID-19 and Individual and Official Actions against the Disease—A Brief Online Survey 6 Months into the Pandemic and on the Threshold of the Second Wave in Germany"

_ijerph, 2021, doi:10.3390/ijerph19010169_

Round 1

Reviewer 1 Report

Dear authors 
I appreciated your attempt to improve statistical methods. Sincerely I had trouble understanding why you used the omnibus test, and you did not report logistic regression results with odds ratio. I ask you to justify your statistical choice in the methods section, providing valuable details about statistical methods used.

Author Response

The reviewer raised an important point and helped us make the results even clearer and more scientific. Therefore, we have sharpened the methods section with the statistical analysis and added another sentence on how to deal with logistic regression analysis (lines 171-173 “The results of the logistic regression analysis as odds ratios are only reported if the omnibus test has previously shown significance for the entire model.“). In addition, we have added the odd's ratio in the results section and no longer reported the omnibus test (lines 264-271).

Reviewer 2 Report

NOTHING

Author Response

Thank you.

This manuscript is a resubmission of an earlier submission. The following is a list of the peer review reports and author responses from that submission.

Round 1

Reviewer 1 Report

Authors already mentioned about the limitation of age of participants like below.  
"The majority of participants were aged between 20 and 30 years and had a higher education status"
However, I recommend enrollment of old age especially over 60 year old, if it is possible. 
Because the high risk group is all age not young age and young age might be not deal this virus as a high pathogen because the mortality rate is very low in young age compare with old age. 
If authors include data of old age, more people may be interested in this manuscript.

Author Response

The reviewer mentioned an important point concerning the underrepresentation of older age groups in our survey. Unfortunately, for our study we won’t be able to include additional older participants because the current situation in the pandemic is different from the situation when we conducted the survey. Knowledge, cases, opinions and vaccine availability have changed and would most likely influence the participants’ responses creating two different data sets. Nevertheless, it is important to intensify the effort to include a higher number of participants from older age groups in future studies.

Reviewer 2 Report

Dear authors

I read with great interest your paper entitled "Synonyms and Symptoms of COVID-19 and Individual and Official Actions against the Disease – A Brief Online Survey 6 Months into the Pandemic and on the Threshold of the Second Wave in Germany".

The topic represents a "hot matter", investigating self-perception of the pandemic according to a specific "keyword" that should be familiar in daily life.

At first, I appreciated your originality to use an online survey.

Here I reported my suggestions.

Introduction: This section is well-written, and the aim of the study is clearly stated: examine the perception of the COVID-19 pandemic by the German population.

Methods: The authors provided a detailed description of how they performed survey and what topic they would investigate. However, there were issues with statistical analysis.

First, due to the high sample size and because the authors stated that they performed subgroups analysis, why did they not perform a statistical comparison, i.e. Chi-square test, to demonstrate if differences were statistically significant?

Second, I suggest that the author perform a logistic regression analysis to find risk factors related to the misperception of relevant aspects of the COVID-19 pandemic. For example, was educational level related to insufficient knowledge of health problems, symptoms or lack of willingness to perform vaccination? In addition, it can be helpful to know if there is a pattern or characteristic related to misperception of the public health matter to adopt specific measures to improve health public vax campaign or "safe" personal behaviour, and so on.  

Results

According to what I reported in the method section, results should be implemented. Moreover, in Figure 2, "Lombardei" is written in the wrong form (correct form "Lombardy"). Therefore, I encourage the authors to examine their data to provide more interesting results.

Discussion

This section should be rewritten according to the results derived from suggestions. 

In conclusion, the topic is fascinating, and I encourage the author to deeply revise their results based on a more robust statistical analysis. For these reasons, the manuscript requires "major revision", and I want to underline that the topic is engaging.

Author Response

Thank you for this important point. We have addressed this aspect by performing Chi-squared tests and logistic regression analysis and included the results in the new version (line 258-273). Therefore we revised the discussion in accordance to the results.

We thank you for the note and changed the spelling in Figure 2 from “Lombardei” to “Lombardy”.

Reviewer 3 Report

Ijerph-1431316-peer_review-v1_DKL_17OCT2021: Synonyms and Symptoms of COVID-19 and Individual and Official Actions against the Disease – A Brief Online Survey 6 Months into the Pandemic and on the Threshold of the Second Wave in Germany

Overall: Well written with clear, concise and impactful readability for numerous sectors; sufficiently researched and analyzed for this type of study; methodological contributions are useful; constraints/limitations recognized; ethics followed; excellent insights and recommendations. Publish as is, or, with minor adjustments if research team/authors think necessary or have any other input they think would be useful. My personal comments/suggestions follow (authors’ choice to include, ignore, adjust… these are merely suggestions, insights and thoughts for future research):

Abstract and Keywords: Clear, good.

Introduction: Good, concise, clear, relevant summary. The methodological problems with phone etc. surveys adequately summarized with subsequent adjustment of methods to compensate (e.g., lines 65-81).

Materials and Methods: Succinctly summarized. It would be interesting for future research to compare Big Data survey/analysis, the approach used for the present paper, and a Deep/Thick Data respondent analysis (semi-structured more intensive interviews; with quasi-bootstrapping by asking how they [respondents] think other segments/demographics of the population parallel or differ from their own attitudes, beliefs, knowledge, behavior, etc.) – this would also provide a comparative methodological analysis useful for current and similar problems and future research. The ethics considerations: Good.

Line 121: “… Then, the questionnaire was…” A bit colloquial; suggested: “… Subsequently, the questionnaire…” or “… The questionnaire was then/subsequently….” (very minor editorial suggestion). Same with line 129. Can also break up long sentences to increase readability.

Example: “The questionnaire was activated from August 6 to August 17, 2020. At first, it was accessible to Prolific users, and 750 complete data records were reached via the platform by August 10 on a first come, first served basis (in total, 760 participants were recruited 128 via the Prolific Platform). Then, the questionnaire was made accessible to recipients of the university’s e-mailing list from August 10 to August 17, which resulted in an additional 153 data records. “

Suggested: ‘The questionnaire was activated from August 6 to August 17, 2020. Initially, it was 126 accessible to Prolific users. 750 complete data records were reached via the platform by August 10 on a first come, first served basis. A total of 760 participants were recruited via the Prolific Platform. In addition, the questionnaire was made accessible to recipients of the university’s e-mailing list from August 10 to August 17. This resulted in an additional 153 data records.’

How was age collected (5 year intervals represented in Figure 1)? I bring this up because many Social Scientists and others erroneously use age brackets (e.g, 10-19; 20-29; 30-39…). This often skews clusters or divides peaks, slopes, troughs, etc. in curves. Asking for exact age is just as easy to collect, analyze and increases accuracy. Often a peak may occur at 18-23 years old, for example (common high school to work or university transition ages) and this is divided rather than accommodated resulting in inaccurate analysis and results. Same with retirement ages and so forth.

Likert scale and check question approaches good/adequate.

Is there a sample questionnaire and summary statistics for answers attached (the figures/tables provide sufficient analyzed data and stats for the research intent, however)?

Adjusted for outliers, incompleteness, inconsistencies, rushed completion, control question failure, etc. (lines 154-162): good.

Good/interesting point (lines 195-198), especially noting that “social-distancing” is not a common phrase in German. Is there something that is used that parallels the concept? Also, what are normative social distance behaviors in Germany (and among rural, urban, age, gender, ethnic, etc. segments)? For example, Cambodians, Midwest Americans, Mongolians, etc. maintain social distance as a norm; Japanese and other Asian groups do not engage in much physical contact in public; while other ethnic/cultural groups have high normative physical contact with hands, bodies, kissing, close facial and bodily engagement in public and so forth. Is/are these considerations relevant factors? Food for thought with further comparative research.

“Vaccine mandate” less importance and greater deviation… Is this a result of a divided social awareness, lack of information, misinformation, polarized beliefs, etc.? For example, the vaccine advocates and the anti-vaccine groups are highly divided/polarized in the US to the point of bullying, threats, protests and violence. Personally, I would rank vaccines as a top priority and many anti-vaccine complainers have ironically been vaccinated for a spectrum of controllable diseases as standard public and personal health measures from infancy to late childhood, and even adulthood (e.g., MMR, Polio, Tetanus, Flu shots and so forth).

Section 3.3: Adequate. Also, would a condensed table be helpful?

Discussion: Just a note for future comparative studies: it would be interesting to compare results over different time periods from this study to present and future, especially during and after spikes, and, various measures for public health (e.g., crowd size control; public event control; lockdowns; social distancing, mask wearing, vaccine records, devices and apps [for example, see Singapore measures]…), public health education and awareness, etc. – a more longitudinal comparative study that is. Also, public policies and information dissemination (corrections of misinformation; compensation of insufficient information, etc.), healthcare and health knowledge access, etc. correlations might prove useful for determine best practices, high risk gaps, and so forth (partially noted in lines 278-290; also 299-304).

Good points in lines 267-273.

Limitations: good, demonstrates good awareness, useful for interpretation by readers and adjusting for future research.

Conclusion: Adequate, good points and recommendations.

Both Discussion and Conclusion can be expanded for future publications – would like to hear more detailed/nuanced evaluations, opinions, speculations and recommendations from the research team (they demonstrate excellent knowledge, professional skills and communicate well – very impactful for all stakeholders and audiences).

References/Citations: Adequate - this is a data-centric and novel/innovative topic and research contribution of particularly high applied relevance to many fields and public health in general. Excessive referencing is unnecessary. 

Author Response

We thank the reviewer for his very relevant comments and have answered them accordingly in the document point by point.
